# Impact of Water Retention Practices in Forests on the Biodiversity of Ground Beetles (Coleoptera: Carabidae)

Emilia Ludwiczak *, Mariusz Nietupski and Agnieszka Kosewska

Department of Entomology, Phytopathology and Molecular Diagnostics,
University of Warmia and Mazury in Olsztyn, 10-719 Olsztyn, Poland
* Correspondence: emilia.ludwiczak@uwm.edu.pl

**Abstract:** This study was carried out in an area covered by the "Increasing retention capacity and prevention of floods and droughts in forest ecosystems in lowland areas" land drainage development project. The aim was to evaluate the impact of transformations in a habitat following the project's implementation on the structure of assemblages of epigeic ground beetles, in the context of the overall trend of decreasing biodiversity. The entomological observations were commenced before launching the planned hydro-technical development at the study area and were repeated 11 years later. During the two years of observations (2008 and 2019), 3311 specimens of ground beetles, representing 89 species, were captured. Water regulation affected the composition of ground beetle assemblages. After the small water retention program had been completed, a quantitative and qualitative decrease in ground beetles was observed. The groundwork for the development caused some disturbances in the habitat, leading to, amongst other things, higher soil moisture, which was conducive to the establishment of ground beetle assemblages other than those observed before the water retention project. After the hydro-technical objects had been constructed, the share of large and small zoophages with higher moisture preferences (hygrophilous species) increased, while the contribution of xerophilous hemizoophages decreased.

**Keywords:** carabid beetles; forest peat bog; species diversity; sustainable water management; water storage; zooindicators



## 1. Introduction

The natural environment is adaptable and undergoes constant transformations. Such transformations are induced by both natural succession and anthropogenic pressure. As our civilization develops, the pressure of anthropogenic stress factors on ecosystems intensifies, directly influencing the rate of ongoing transformation [1–3]. Sustainable management in ecosystems is a very important issue, especially concerning the maintenance and protection of biodiversity.

Hydrotechnical developments carried out on a large scale play a significant role in anthropogenic transformations of ecosystems [4,5]. Retention works, a symbol of modern agriculture and forestry in the 20th century, contributed to a growth in the total area of land available for economic purposes [6]. However, unidirectional development projects, mainly consisting of the drainage of territories with a high level of surface and ground waters, caused excessive drying of wetlands and peat bogs. The economic aspect of such drainage works wholly overshadowed the natural context, leading to the substantial impoverishment of flora and fauna in drained areas [7]. Poland is among those countries with poor water resources, and this is a particularly acute problem in the forests, which make up 29.6% of the country's total area [8,9]. Boczoń et al. [10] confirm in their study that there was a water deficit in nearly all forests in Poland in 2015. Therefore, the previous problems with water retention, arising from the draining of excess water to create new agricultural land, have been reconsidered. Today's priority is to retain water in natural ecosystems [6,11,12].

The biodiversity of the natural environment is strongly dependent on natural aspects [13] and several anthropogenic factors [14]. Sushko [15] states that even minor disturbances, either natural or man-made, have a negative effect on the presence of insects. Obidziński [16] defines the resulting disturbances as changes occurring in the environment that interfere with the structure and processes in an ecosystem, ultimately destroying the ecosystem's original structure. The unpredictability and volatility of these elements make it very difficult to foresee the direction and pace of change unambiguously [17]. Chapin et al. [18] and Schindler et al. [19] emphasize that natural disturbances such as floods, fires, and hurricanes cannot be viewed as exclusively negative events. The resulting transformations shape the structures of biocenoses and the consequent environmental conditions create new, often valuable populations [20,21]. Although such transformations may cause the disappearance of local species, they are transient and relatively common, and the environment is naturally adapted to such changes [22]. Disturbances induced by anthropopressure, however, frequently cause more severe and lasting changes. Human activity, depending on its intensity, can lead to the loss of characteristics of an ecosystem, both locally and globally. Subsequently, it can cause the collapse of the ecosystem due to the loss of diversity of life forms [23]. The rapid growth of anthropopressure in the 21st century has resulted in biodiversity loss at an unprecedented pace [14,24]. Such transformations have caused a decline in the number, diversity and biomass of insects worldwide [25,26].

The growing awareness that we are responsible for the global decrease in the abundance of insects has stimulated the launch of actions that could help satisfy our needs while simultaneously protecting biodiversity [27]. In this context, an important project undertaken by the State Forests in Poland is small water retention. Because of the widespread water deficit, the aim is to shape the water balance of the whole environment while supporting and protecting biodiversity [28]. Small water reservoirs created as part of the hydroengineering works play an essential role in these processes [4]. The influence of small water retention basins on shaping the food base for ground beetles, thus counteracting their qualitative and quantitative regression [29], has been demonstrated, for example, by Czyżyk and Porter [30]. Anthropopressure is unavoidable, but sustainable management of landscape resources can guide us towards correct solutions, which consider both economic and environmental aspects [31]. The "Increasing retention capacity and prevention of floods and droughts in forest ecosystems in lowland areas" project, referred to as "small retention", is being implemented in the State Forest Farms of Poland's State Forests (PGL LP) to improve the moisture content [32–34]. Within this project, the Dąbrówka Forest Subdistrict was chosen in 2007 as a site for implementing the EU-co-funded program "A small retention program for the Warmińsko-Mazurskie voivodeship in 2006–2015" [12]. This project's stabilization of water relations is mainly achieved through rational forest management methods, includincludeding technical and non-technical measures [6,35]. Launching and implementing a small retention program entails some disturbances in a given area, which directly impact the local flora and fauna. The intensity of this impact, its duration, and the ultimate effect of transformations caused in the habitat require reliable assessment. Bioindication methods, which are the fundamental approach in assessing ongoing environmental changes, are a standard tool employed in such evaluation [36,37]. Beetles from the Carabidae family are a group of bioindicators frequently used in monitoring the environment, as confirmed by Pearce and Venier [38], Avgin and Luff [39], and Langraf et al. [40]. The presence of epigeic fauna composed of ground beetles in nature depends on their habitat, the availability of food resources, and meteorological conditions. However, the critical factor strongly correlating with their basic life functions is water [41]. This issue has been verified empirically by Luff [42], who maintains that a habitat's humidity is the overriding ecological factor in the life of insects. Ground beetles are susceptible to all anthropogenic transformations, especially those which modify water management [40,43,44]. Therefore, assemblages of Carabidae reactions at the morphological, anatomical, and behavioral levels, reflect the general interactions taking place in the transformed environment [45]. The following assessment of the environmental trans-

formation pressure on the fauna of ground beetles was based on changes in the species composition and structure of Carabidae assemblages in the area of the Dąbrówka Forest Subdistrict. Based on previous research carried out in this area [46] and on the basis that biodiversity diminishes as water availability decreases [44,47], several working hypotheses were proposed:

- implementation of hydro-technical modifications in the analyzed forest subdistrict will increase the abundance and species diversity of Carabidae and changes shares of ecological groups of Carabidae;
- improved moisture relations in the habitat, in the long term, will induce changes in the composition of ground beetles by raising the share of valuable hygrophilous fauna;
- lasting modifications to the water relations in the habitat will lead to changes in the carabid beetle assemblages caused by changes in dominant species.

## 2. Materials and Methods

### 2.1. Study Area

This study on the epigeic fauna composed of ground beetles was carried out in the area of the Dąbrówka Forest Subdistrict (north-eastern part of Poland, UTM DE 66).

The water retention project in the Dąbrówka Forest Subdistrict was launched in 2010 by first making a plan (including verifying the local environmental conditions and obtaining all necessary permits), followed by engineering works in 2011 and 2012. The main objective was to construct water retention reservoirs, which would collect water during floods and store it until needed during dry weather [28].

The experiment comprised two research (sampling) sites located at different distances from the two retention reservoirs (a, b) built under the project (each retention basin was 0.07 ha in area) [48] (Figure 1). Similar habitat conditions characterized both research sites. Both were surrounded by marshy mixed forest with peat plant cover, with willow (*Salix alba* L.) being the dominant plant in the undergrowth. The whole habitat lies on the peat soils of a transitional peat bog with a high moisture content [48]. This area was adjacent to a 21-year-old fresh mixed forest (covering 18.28 ha to the east and 2.93 ha to the west of the research sites). In addition to the dominant Scots pine (*Pinus sylvestris* L.), the tree stand was composed of warty birch (*Betula pendula* Roth), European larch (*Larix decidua* Mill.), black alder (*Alnus glutinosa* Gaertn.), small-leaved linden (*Tilia cordata* Mill.), field elm (*Ulmus minor* Mill.), and Norway spruce (*Picea abies* (L.) H. Karst) [48].

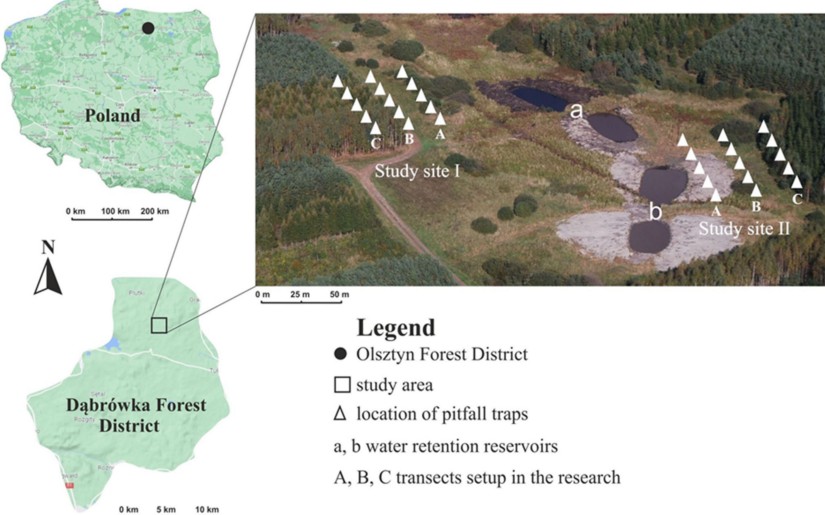

**Figure 1.** Approximate location of the study sites (I, II), water retention reservoirs (a, b) and the transects (A–C). Source: www.google.pl/maps (accessed on 27 September 2022) [49], Olsztyn Forest District.

One research site was located at a distance of 128 m from a dirt road, 36 m from retention basin a, and 62 m from retention basin b. The other site, situated to the east of the retention basins, lies at a distance of 100 m from the first site, 47 m from the dirt road, 39 m from the water retention basin a, and 15 m from water retention basin b [48]. Three transects (zones) were set out linearly in each sampling site. Transect A, located on a wet bog meadow, crossed the area with the highest moisture content. Transect B was 20 m away from transect A and was in the ecotone zone between an extensive meadow and a mixed coniferous forest. This transect was characterized by lower water availability in terms of water content. The third transect (C) ran through the forest and was distinguished by the lowest water content. It lay 20 m from transect B (Figure 1).

Measurements of soil moisture in 2008 and 2019 were made using an electronic soil moisture meter of the LB-797 type with a probe equipped with two needle electrodes. In each year, the humidity was tested four times (June, July, August, September). In the area of 0.5 m$^2$ around each trap, 5 readings of the value of the tested indicator were made, from which the average was then calculated (Table 1).

**Table 1.** Medium humidity levels in the analyzed area before (2008) and after (2019_1; 2019_2) the implementation of the small water retention project.

| Humidity (%) | | | | | | | | | | | | | | |
|---|---|---|---|---|---|---|---|---|---|---|---|---|---|---|
| | | | | | | | 2008 | | | | | | | |
| | 5.05[1] | | | 2.06 | | | 17.07 | | | 11.08 | | | 8.09 | |
| A | B | C | A | B | C | A | B | C | A | B | C | A | B | C |
| 36.8 | 21.5 | 14.8 | 30.5 | 11.8 | 9.5 | 33.6 | 13.4 | 12.8 | 25.9 | 14.3 | 10.4 | 33.1 | 14.2 | 11.6 |
| | | | | | | | 2019_1 | | | | | | | |
| | 8.05 | | | 5.06 | | | 19.07 | | | 14.08 | | | 11.09 | |
| A | B | C | A | B | C | A | B | C | A | B | C | A | B | C |
| 33.9 | 19.0 | 12.8 | 33.2 | 26.0 | 17.7 | 33.2 | 21.0 | 15.7 | 36.0 | 27.2 | 22.8 | 33.1 | 22.1 | 17.1 |
| | | | | | | | 2019_2 | | | | | | | |
| | 8.05 | | | 5.06 | | | 19.07 | | | 14.08 | | | 11.09 | |
| A | B | C | A | B | C | A | B | C | A | B | C | A | B | C |
| 23.8 | 22.1 | 13.8 | 32.9 | 31.0 | 20.0 | 28.0 | 30.6 | 16.0 | 36.1 | 38.3 | 23.0 | 30.5 | 30.1 | 16.2 |

[1] Date of observation.

### 2.2. Data Collection

Our study on the epigeic Carabidae fauna started on research site I in 2008. The main purpose was to evaluate the initial condition of the natural environment in the area where the hydro-technical development was to be implemented. In each transect (A, B, and C), five soil traps were installed at a 10-m distance from each other (1 research site × 3 transects × 5 traps). In 2019, the entomological observations were expanded by setting out another research site, II, situated to the east of the water retention basins (2 research sites: 2019_1, 2019_2 × 3 transects × 5 traps) (Figure 1). One trap from the analyzed set located in a given research site was treated as a replication. In the first research year (2008), carabids were captured from 20 April to 3 November (197 days), whereas in 2019, the entomological observations were carried out from 10 May to 22 November (196 days). Modified soil traps were used to catch beetles. They consisted of plastic containers (500 mL in capacity, 12 cm in height) filled up to 1/3 with ethylene glycol solution. Soil traps were dug into the soil, level with the ground's surface, and emptied every 14 days.

### 2.3. Data Analysis

Our study used a quantitative method expressing the number of caught specimens and species. The ecological analysis relied on basic parameters, such as the body size, development strategy, feeding preferences, preferred habitat and moisture conditions, and ability to fly. The following division was applied: trophic group—large zoophages (Lz) (>15 mm body length), small zoophages (Sz) (<15 mm), hemizoophages (Hz); phenology—

spring breeders (sb), autumn breeders (ab); hygropreferences—hygrophilous species (H), mesophilous species (M), xerophilous species (Xe); habitat types—forest species (F), open area species (Oa), generalists (G) and wetland species (We) [50–55]. The examined beetles were also classified according to their dispersion capability. According to the description of Hůrka [56], three groups were distinguished: macropterous (fully developed wings), brachypterous (reduced second pair wings), and dipterous (second pair wings can be developed or reduced). The distribution of means of the number of specimens, species, and MIB (Mean Individual Biomass) was assessed [57] and the abundance of the various ecological groups of carabids, was measured using the Shapiro–Wilk test. The significance of differences was evaluated using the Generalized Linear Model (GML). The Jackknife 2 estimator was applied to abundance data (using EstimateS v. 9.1.0 statistical software), and species accumulation curves were calculated to assess the adequacy of the sampling efficiency [58,59].

Differences between the analyzed ground beetle assemblages in the two study years and the transects are presented with the help of non-metric multidimensional scaling (NMDS), using Morisita's measure of similarity. This method enabled us to evaluate assemblages of carabid beetles based on the similarity between pairs of samples while analyzing the data on the number of captured species and individuals. Samples showing a high degree of similarity are placed close to each other in the diagram. The significance of differences between the insect assemblages with the NMDS method was assessed using the ANOSIM non-parametric statistical test [60,61].

Our analysis of the dependences between the presence of the identified ecological groups of Carabidae in the two years studied followed ordination techniques [62]; i.e., Canonical Correspondence Analysis (CCA) was used. The distinguishing factors were the two years of observations and the change in the environment (moisture content), and their significance was evaluated using the Monte Carlo permutation test.

All statistical computations and their visualization were carried out using the following software packages: Statistica 13.3, PAST, and Canoco 4.51.

## 3. Results

During the two years of observations (2008 and 2019), 3331 specimens of ground beetles, representing 89 species, were captured. In the first year, 2085 specimens from 73 species were caught. After the hydro-technical changes were implemented, 1226 specimens in total from both research sites (668 from site I and 658 from site II) classified into 58 species (47 in site I and 44 in site II) were collected (Table 2).

**Table 2.** Species composition, number of specimens and ecological description of Carabidae caught in the analyzed area before (2008) and after (2019_1; 2019_2) the implementation of the small water retention project.

| Species | Ecological Description * | 2008 | | | 2019_1 | | | 2019_2 | | |
|---|---|---|---|---|---|---|---|---|---|---|
| | | A ** | B | C | A | B | C | A | B | C |
| *Acupalpus exiguus* Dejean, 1829 | We, H, Hz, Sb | 1 | 0 | 0 | 0 | 0 | 0 | 0 | 0 | 0 |
| *Acupalpus flavicollis* (J. Sturm, 1825) | We, H, Hz, Sb | 0 | 0 | 0 | 0 | 0 | 0 | 1 | 0 | 0 |
| *Agonum fuliginosum* (Panzer, 1809) | We, H, Sz, Sb | 21 | 1 | 0 | 0 | 1 | 0 | 2 | 2 | 0 |
| *Amara aenea* (De Geer, 1774) | Oa, Xe, Hz, Sb | 18 | 23 | 22 | 0 | 1 | 0 | 1 | 0 | 0 |
| *Amara aulica* (Panzer,1797) | Oa, M, Hz, Ab | 2 | 0 | 0 | 0 | 0 | 0 | 0 | 0 | 0 |
| *Amara bifrons* (Gyllenhal, 1810) | Oa, Xe, Hz, Ab | 10 | 14 | 10 | 0 | 0 | 1 | 0 | 0 | 0 |
| *Amara brunnea* (Gyllenhal, 1810) | F, M, Hz, Ab | 0 | 0 | 3 | 3 | 0 | 7 | 1 | 3 | 24 |
| *Amara communis* (Panzer, 1797) | Oa, M, Hz, Sb | 30 | 10 | 4 | 1 | 12 | 14 | 2 | 6 | 1 |
| *Amara consularis* (Duftschmid,1812) | Oa, Xe, Hz, Ab | 0 | 2 | 1 | 0 | 0 | 0 | 0 | 0 | 0 |
| *Amara convexior* Stephens, 1828 | Oa, M, Hz, Sb | 22 | 14 | 2 | 1 | 3 | 0 | 0 | 0 | 0 |
| *Amara curta* Dejean, 1828 | Oa, Xe, Hz, Sb | 1 | 3 | 0 | 0 | 0 | 0 | 0 | 0 | 0 |
| *Amara equestris* (Duftschmid, 1812) | Oa, Xe, Hz, Ab | 0 | 2 | 0 | 0 | 1 | 0 | 0 | 0 | 0 |
| *Amara eurynota* (Panzer,1797) | Oa, M, Hz, Ab | 0 | 1 | 0 | 0 | 0 | 0 | 0 | 0 | 0 |

**Table 2.** *Cont.*

| Species | Ecological Description * | 2008 | | | 2019_1 | | | 2019_2 | | |
|---|---|---|---|---|---|---|---|---|---|---|
| | | A ** | B | C | A | B | C | A | B | C |
| *Amara familiaris* (Duftschmid, 1812) | G, M, Hz, Sb | 2 | 0 | 0 | 1 | 0 | 0 | 0 | 0 | 0 |
| *Amara fulva* (Degeer,1774) | Oa, M, Hz, Ab | 0 | 1 | 1 | 0 | 0 | 0 | 0 | 0 | 0 |
| *Amara lunicollis* Schiodte, 1837 | Oa, M, Hz, Sb | 13 | 29 | 5 | 0 | 1 | 0 | 0 | 0 | 0 |
| *Amara municipalis* (Duftschmid,1812) | Oa, M, Hz, Ab | 1 | 4 | 3 | 0 | 0 | 0 | 0 | 0 | 0 |
| *Amara similata* (Gyllenhal, 1810) | Oa, M, Hz, Sb | 1 | 0 | 0 | 0 | 0 | 0 | 0 | 0 | 0 |
| *Amara spreta* Dejean, 1831 | Oa, Xe, Hz, Sb | 1 | 7 | 1 | 0 | 0 | 0 | 0 | 0 | 0 |
| *Amara tibialis* (Paykull, 1798) | G, Xe, Hz, Sb | 0 | 4 | 1 | 0 | 0 | 0 | 0 | 0 | 0 |
| *Anisodactylus binotatus* (Fabricius, 1787) | G, H, Hz, Sb | 0 | 0 | 0 | 1 | 0 | 0 | 1 | 0 | 0 |
| *Badister bullatus* (Schrank,1798) | G, M, Sz, Sb | 0 | 0 | 0 | 1 | 1 | 0 | 2 | 1 | 0 |
| *Badister lacertosus* Sturm,1815 | F, M, Sz, Sb | 0 | 0 | 0 | 0 | 0 | 0 | 2 | 0 | 0 |
| *Badister sodalis* (Duftschmid,1812) | We, H, Sz, Sb | 0 | 0 | 0 | 0 | 0 | 0 | 1 | 0 | 0 |
| *Bembidion gilvipes* (Sturm, 1825) | Oa, H, Sz, Sb | 42 | 1 | 0 | 4 | 0 | 0 | 4 | 2 | 0 |
| *Bembidion guttula* (Fabricius,1792) | Oa, H, Sz, Sb | 2 | 0 | 0 | 0 | 0 | 0 | 1 | 0 | 0 |
| *Bembidion mannerheimii* (C.Sahlberg,1827) | We, H, Sz, Sb | 4 | 0 | 0 | 0 | 0 | 0 | 0 | 0 | 0 |
| *Blemus discus* (Fabricius,1792) | We, M, Sz, Ab | 0 | 0 | 0 | 0 | 0 | 0 | 1 | 0 | 0 |
| *Bradycellus csikii* Laczo,1912 | Oa, M, Hz, Sb | 3 | 0 | 2 | 0 | 0 | 0 | 0 | 0 | 0 |
| *Bradycellus harpalinus* (Audinet-Serville,1821) | Oa, Xe, Hz, Sb | 3 | 4 | 1 | 0 | 1 | 0 | 0 | 1 | 0 |
| *Calathus ambiguus* (Paykull, 1790) | Oa, Xe, Sz, Ab | 1 | 0 | 0 | 0 | 0 | 0 | 0 | 0 | 0 |
| *Calathus erratus* (C.Sahlberg, 1827) | G, Xe, Sz, Ab | 5 | 192 | 60 | 0 | 0 | 1 | 0 | 0 | 1 |
| *Calathus fuscipes* Goeze, 1777 | Oa, M, Sz, Ab | 1 | 1 | 1 | 0 | 0 | 0 | 0 | 0 | 0 |
| *Calathus melanocephalus* (Linnaeus, 1758) | Oa, M, Sz, Ab | 7 | 7 | 2 | 0 | 0 | 0 | 0 | 0 | 0 |
| *Calathus micropterus* (Duftschmid, 1812) | F, M, Sz, Ab | 0 | 0 | 0 | 1 | 4 | 3 | 0 | 2 | 24 |
| *Carabus cancellatus* Illiger,1798 | G, M, Lz, Sb | 1 | 0 | 0 | 0 | 0 | 0 | 0 | 0 | 0 |
| *Carabus convexus* Fabricius,1775 | F, Xe, Lz, Sb | 1 | 2 | 0 | 0 | 0 | 0 | 0 | 2 | 0 |
| *Carabus glabratus* Paykull, 1790 | F, M, Lz, Ab | 0 | 1 | 0 | 2 | 1 | 2 | 2 | 0 | 2 |
| *Carabus granulatus* (Linnaeus, 1758) | We, H, Lz, Sb | 2 | 0 | 0 | 8 | 3 | 3 | 4 | 4 | 4 |
| *Carabus hortensis* Linnaeus, 1758 | F, M, Lz, Ab | 0 | 2 | 4 | 8 | 1 | 21 | 7 | 10 | 73 |
| *Carabus marginalis* Fabricius, 1794 | F, M, Lz, Sb | 36 | 38 | 27 | 7 | 10 | 12 | 2 | 4 | 15 |
| *Carabus nemoralis* O.F. Müller, 1764 | G, M, Lz, Sb | 0 | 4 | 8 | 0 | 0 | 6 | 0 | 0 | 5 |
| *Carabus violaceus* Linnaeus, 1758 | F, M, Lz, Ab | 0 | 0 | 0 | 0 | 0 | 1 | 0 | 0 | 0 |
| *Cicindela campestris* Linnaeus, 1758 | Oa, Xe, Lz, Sb | 1 | 0 | 0 | 0 | 1 | 0 | 0 | 0 | 0 |
| *Clivina fossor* (Linnaeus, 1758) | Oa, M, Sz, Sb | 15 | 0 | 0 | 7 | 3 | 0 | 11 | 5 | 0 |
| *Dyschirius globosus* (Herbst, 1784) | Oa, H, Sz, Sb | 89 | 4 | 2 | 27 | 8 | 15 | 65 | 47 | 1 |
| *Elaphrus cupreus* Duftschmid, 1812 | We, H, Sz, Sb | 1 | 0 | 0 | 0 | 0 | 0 | 0 | 0 | 0 |
| *Epaphius rivularis* (Schrank, 1781) | We, H, Sz, Ab | 6 | 0 | 0 | 0 | 0 | 0 | 0 | 0 | 0 |
| *Epaphius secalis* (Paykull, 1790) | G, M, Sz, Ab | 49 | 15 | 1 | 24 | 26 | 10 | 5 | 12 | 1 |
| *Harpalus affinis* (Schrank, 1781) | G, Xe, Hz, Sb | 4 | 1 | 1 | 0 | 0 | 0 | 0 | 0 | 0 |
| *Harpalus anxius* (Duftschmid, 1812) | Oa, Xe, Hz, Sb | 0 | 0 | 1 | 0 | 0 | 0 | 0 | 0 | 0 |
| *Harpalus griseus* (Panzer, 1796) | Oa, Xe, Hz, Ab | 0 | 2 | 1 | 0 | 0 | 0 | 0 | 0 | 0 |
| *Harpalus laevipes* Zetterstedt, 1828 | F, M, Hz, Sb | 0 | 0 | 0 | 0 | 0 | 0 | 0 | 0 | 1 |
| *Harpalus latus* (Linnaeus, 1758) | G, M, Hz, Ab | 3 | 13 | 0 | 0 | 3 | 0 | 0 | 0 | 0 |
| *Harpalus luteicornis* (Duftschmid, 1812) | Oa, M, Hz, Sb | 1 | 0 | 0 | 0 | 0 | 0 | 0 | 0 | 0 |
| *Harpalus rubripes* (Duftschmid, 1812) | Oa, Xe, Hz, Sb | 14 | 34 | 5 | 0 | 0 | 0 | 0 | 0 | 0 |
| *Harpalus rufipalpis* J. Sturm, 1818 | Oa, Xe, Hz, Sb | 1 | 4 | 2 | 0 | 0 | 0 | 0 | 0 | 0 |
| *Harpalus rufipes* (De Geer, 1774) | Oa, M, Hz, Ab | 21 | 35 | 17 | 0 | 1 | 0 | 1 | 0 | 0 |
| *Harpalus signaticornis* (Duftschmid, 1812) | Oa, Xe, Hz, Sb | 0 | 0 | 4 | 0 | 0 | 0 | 0 | 0 | 0 |
| *Harpalus smaragdinus* (Duftschmid, 1812) | Oa, Xe, Hz, Sb | 0 | 5 | 7 | 0 | 0 | 0 | 0 | 0 | 0 |
| *Harpalus tardus* (Panzer, 1796) | Oa, Xe, Hz, Sb | 65 | 109 | 82 | 0 | 1 | 0 | 0 | 0 | 0 |
| *Leistus ferrugineus* Linnaeus, 1758 | F, M, Sz, Ab | 0 | 1 | 0 | 0 | 0 | 1 | 0 | 0 | 1 |
| *Leistus rufomarginatus* (Duftschmid, 1812) | F, M, Sz, Ab | 0 | 0 | 0 | 0 | 0 | 0 | 0 | 0 | 1 |
| *Leistus terminatus* Panzer, 1793 | We, H, Sz, Ab | 1 | 2 | 2 | 1 | 0 | 3 | 0 | 1 | 0 |
| *Limodromus assimilis* Paykull, 1790 | F, H, Sz, Sb | 0 | 0 | 0 | 0 | 0 | 0 | 0 | 0 | 1 |
| *Loricera pilicornis* (Fabricius, 1775) | We, H, Sz, Sb | 5 | 0 | 0 | 0 | 0 | 0 | 0 | 0 | 0 |
| *Microlestes minutulus* (Goeze, 1777) | Oa, Xe, Sz, Ab | 2 | 1 | 0 | 1 | 1 | 0 | 0 | 3 | 0 |
| *Nebria brevicollis* (Fabricius, 1792) | G, M, Sz, Ab | 0 | 0 | 0 | 1 | 0 | 0 | 0 | 1 | 3 |
| *Notiophilus palustris* (Duftschmid, 1812) | G, M, Sz, Sb | 0 | 1 | 0 | 0 | 0 | 1 | 0 | 0 | 0 |

**Table 2.** *Cont.*

| Species | Ecological | 2008 | | | 2019_1 | | | 2019_2 | | |
|---|---|---|---|---|---|---|---|---|---|---|
| | Description * | A ** | B | C | A | B | C | A | B | C |
| *Oodes helopioides* (Fabricius, 1792) | We, H, Sz, Sb | 1 | 0 | 0 | 0 | 0 | 0 | 2 | 0 | 0 |
| *Oxypselaphus obscurus* (Herbst, 1784) | F, H, Sz, Sb | 3 | 0 | 0 | 2 | 1 | 1 | 5 | 2 | 1 |
| *Patrobus atrorufus* (Str?m, 1768) | We, H, Sz, Ab | 3 | 1 | 0 | 0 | 0 | 0 | 0 | 0 | 0 |
| *Philorhizus sigma* (P. Rossi, 1790) | Oa, Xe, Sz, Sb | 0 | 0 | 0 | 0 | 1 | 0 | 0 | 0 | 0 |
| *Poecilus cupreus* (Linnaeus, 1758) | Oa, M, Sz, Sb | 2 | 2 | 1 | 0 | 0 | 0 | 0 | 0 | 0 |
| *Poecilus lepidus* (Leske, 1785) | Oa, Xe, Sz, Sb | 3 | 26 | 6 | 0 | 0 | 0 | 0 | 0 | 0 |
| *Poecilus versicolor* (J. Sturm, 1824) | G, M, Sz, Sb | 2 | 3 | 0 | 2 | 0 | 0 | 2 | 1 | 0 |
| *Pterostichus aterrimus* (Herbst, 1784) | We, H, Lz, Sb | 0 | 0 | 0 | 2 | 0 | 0 | 0 | 0 | 0 |
| *Pterostichus diligens* (Sturm, 1824) | We, H, Sz, Sb | 26 | 5 | 0 | 3 | 0 | 0 | 0 | 2 | 0 |
| *Pterostichus melanarius* (Illiger,1798) | G, M, Lz, Sb | 0 | 0 | 0 | 0 | 1 | 0 | 1 | 0 | 1 |
| *Pterostichus minor* (Gyllenhal, 1827) | We, H, Sz, Sb | 1 | 1 | 0 | 1 | 0 | 0 | 1 | 0 | 0 |
| *Pterostichus niger* (Schaller, 1783) | F, M, Lz, Ab | 150 | 140 | 134 | 39 | 86 | 94 | 21 | 53 | 124 |
| *Pterostichus nigrita* (Paykull, 1790) | We, H, Sz, Sb | 0 | 1 | 0 | 3 | 8 | 0 | 0 | 0 | 0 |
| *Pterostichus oblongopunctatus* (Fabricius, 1787) | F, M, Sz, Sb | 0 | 0 | 1 | 2 | 0 | 5 | 0 | 2 | 32 |
| *Pterostichus strenuus* (Panzer, 1796) | G, H, Sz, Sb | 16 | 3 | 0 | 1 | 0 | 3 | 3 | 1 | 2 |
| *Pterostichus vernalis* (Panzer, 1796) | G, M, Sz, Sb | 63 | 67 | 22 | 8 | 16 | 2 | 3 | 13 | 3 |
| *Stenolophus mixtus* (Herbst, 1784) | Oa, H, Sz, Sb | 1 | 0 | 0 | 0 | 0 | 0 | 0 | 0 | 0 |
| *Stomis pumicatus* (Panzer,1796) | G, M, Sz, Ab | 0 | 0 | 0 | 0 | 0 | 0 | 1 | 0 | 0 |
| *Syntomus truncatellus* (Linne, 1761) | Oa, Xe, Sz, Sb | 5 | 2 | 1 | 0 | 1 | 0 | 2 | 0 | 0 |
| *Synuchus vivalis* (Illiger, 1798) | Oa, Xe, Sz, Ab | 4 | 2 | 1 | 0 | 1 | 1 | 0 | 0 | 0 |
| Individuals total | | 789 | 847 | 449 | 162 | 199 | 207 | 157 | 180 | 321 |
| | | | | | | 3 311 | | | | |
| Number of species | | 56 | 50 | 38 | 28 | 29 | 22 | 30 | 24 | 22 |
| | | | | | | 89 | | | | |

* Lz—large zoophages, Sz—small zoophages, Hz—hemizoophages, Sb—spring breeders, Ab—autumn breeders, H—hygrophilous, M—mesophilous, Xe—xerophilous, F—forest, Oa—open area, G—generalists, We—wetland; **—transects.

*Pterostichus niger*, a woodland species with moderate moisture requirements, was the prevalent species in both years. Before the retention project was launched, the other dominant species were xerophilous *Calathus erratus* and xerophilous, open-area species *Harpalus tardus*. In 2019, the following species were caught in large numbers: *Dyschirius globosus*, a hygrophilous species dwelling in open areas, and mesophilous species *Carabus hortensis* and *Epaphius secalis* (Table 2). In the second year (2019), a decreasing share of species with low moisture requirements was recorded. Only single specimens of the xerophilous species *C. erratus*, *H. tardus*, *Amara aenea*, *Amara bifrons*, *Syntomus truncatellus* and *Bradycellus harpalinus* were caught. Additionally, representatives of the xerophilous species *Harpalus rubripes* and *Poecilus lepidus* were unobserved (Table 2).

The species accumulation curves for each transect in both study years confirmed adequate sampling effort (Figure 2). Only the rarefaction curve in the ecotone zone (B) in sampling site I in the second year of the study failed to reach an asymptote (Figure 2).

Statistical analysis was carried out to determine the significance of differences between the average values of the abundance, species composition, and mean individual biomass (MIB) in each site and study year. The compliance of the data obtained in our experiment with normal distribution was tested using the Shapiro–Wilk test, which revealed a unimodal distribution. A generalized linear model (GLM) analysis was conducted in the absence of normal distribution, which enabled us to determine *p* values (Table 3). A significant effect of the analyzed parameters in both study years and sites on the number and mean individual biomass of ground beetles was determined. The species richness was significantly affected by the year, whereas the impact of the research object proved to be an insignificant factor (Table 3).

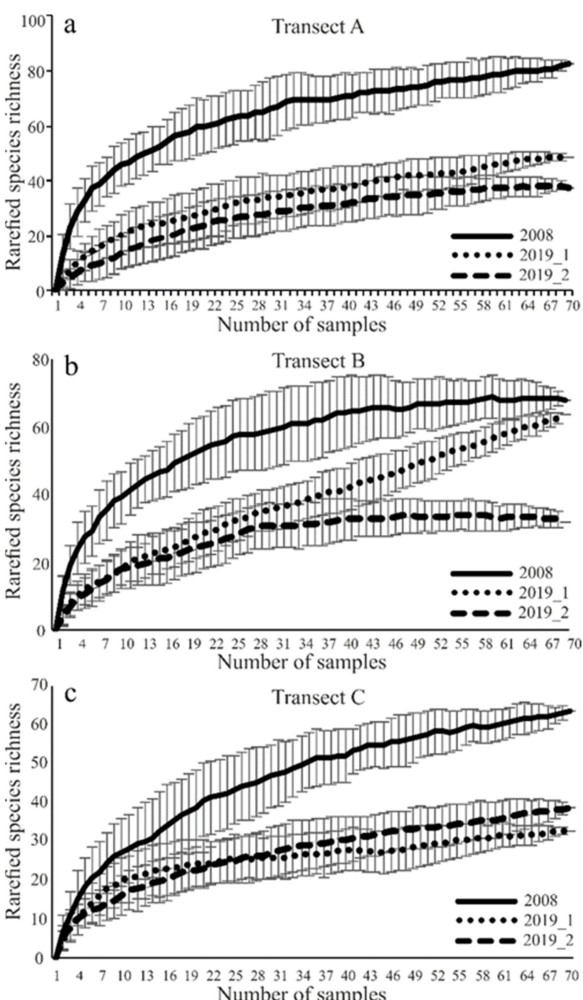

**Figure 2.** (**a**)rarefied species richness in transect A; (**b**): rarefied species richness in transect B; (**c**) rarefied species richness in transect C; Expected number of carabid species caught in the studied sites using the Jackknife estimator (±SD) of species richness (species accumulation curves for carabid beetles assemblages sampled in three transects (A, B, C) in years of study (2008, 2019), using the Jackknife).

**Table 3.** Results of the generalized linear model (GLM) test of significance in terms of the abundance, species richness, and mean individual biomass (MIB) of ground beetles in the analyzed area. Tests for the significance of the effects in the model were done using the Wald's statistics.

|  | **Wald's Statistics** | *p* |
|---|---|---|
| Individuals |  |  |
| Area | 8.5 | 0.01 |
| Year | 1197.82 | 0.00 |
| Species |  |  |
| Area | 1.02 | 0.6 |
| Year | 340.18 | 0.00 |
| MIB |  |  |
| Area | 8920.00 | 0.00 |
| Year | 1193.00 | 0.00 |

The average species richness, abundance, and MIB values of the carabid fauna captured in each study year and transect were varied (Figure 3a–c. However, both before and following the construction of the retention basins (in 2008 and 2019, respectively), the entomological material in transects A and B was characterized by similar values of the parameters (Figure 3a–c). Significant differences were observed in the material collected along the transect crossing the forest (C). In 2008, this transect represented the lowest number and species richness of ground beetles. In contrast, in 2019 the number of specimens and the richness of species of these beetles in sampling site II were higher than along the other transects. Different dependencies were observed when studying carabid beetles' mean individual body weight (MIB). In both study years, higher parameter values were observed in forest transect C. In contrast, the lowest ones were determined along transect A, corresponding to the area with the highest moisture content (Figure 3a–c).

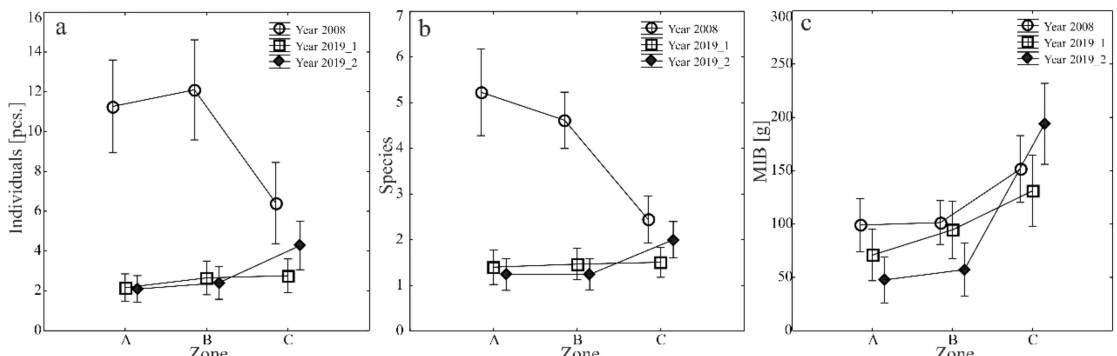

**Figure 3.** (**a**–**c**) Mean numbers of specimens (**a**), species (**b**) and MIB (**c**) of Carabidae caught in the analyzed area in 2008–2019 (vertical lines indicate SE).

The NMDS analysis (ANOSIM; A: R = 0.77, B: R = 0.90, C: R = 0.63; *p* < 0.0001), through the absence of any overlap of the polygons, confirmed the occurrence of a significant distinction between Carabidae assemblages in the two study years (NMDS—zone A—a, zone B—b and zone C—c) (Figure 4a–c. Moreover, the curves plotted in the diagrams to visualize the differences, and their statistical interpretation, revealed the highest degree of separation in transects A and B, as well as considerable differences in the composition of Carabidae assemblages within transect C in 2008 (Stress A = 0.1, B = 0.09, C = 0.1) (Figure 4a–c).

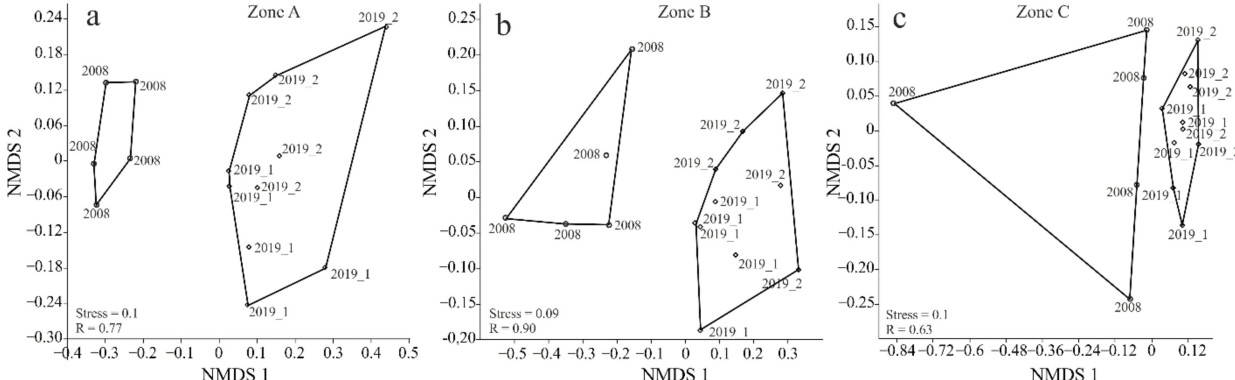

**Figure 4.** Diagram of nonmetric multivariate scaling (NMDS) for the Carabidae ground clusters of the studied sites in the years of research (2008–2019 zone A—(**a**), zone B—(**b**) and zone C—(**c**)).

The outcome of the generalized linear model (GLM) for the abundance of the analyzed ecological groups, including the variables (zones, study years), showed significant differences for all ecological groups (Table 4).

**Table 4.** Results of the generalized linear model (GLM) for the abundance of carabid ecological groups in the analyzed zones in the two study years (2008, 2019). Tests for the significance of the effects in the model were done using the Wald's statistics.

|  | **Wald's Statistics** | *p* |
|---|---|---|
| Eu * | 132.34 | 0.00 |
| F | 448.42 | 0.00 |
| Oa | 980.96 | 0.00 |
| We | 94.03 | 0.00 |
| H | 426.44 | 0.00 |
| M | 477.62 | 0.00 |
| Xe | 567.87 | 0.00 |
| Hz | 646.90 | 0.00 |
| Lz | 206.47 | 0.00 |
| Sz | 692.44 | 0.00 |
| Ab | 542.99 | 0.00 |
| Sb | 996.49 | 0.00 |

*—all abbreviations are as given in the methodology section.

The ecological characteristics of the captured ground beetles showed an increase in the share of woodland species (F) and wetland species (We) when the water retention project was completed (Figure 5a–d). In both study years, the entomological material was dominated by mesophilous carabid beetles, with a broad spectrum of moisture preferences (M) (Figure 6a–d). In both study years, small zoophages (Sz) dominated transects A and B, while a prevalence of large zoophages was recorded in transect C. Meanwhile, the analysis of the 2019 data demonstrated a substantial decline in the share of hemizoophages (Hz) and carabids with low moisture requirements (Xe) (Figure 7a–c). The phenological analysis of the three transects set out in this experiment in both study years showed the dominance of ground beetles of the spring type of development (Sb) in transect A and the dominance of typically autumnal species (Ab) in transects B and C (Figure 8a, b). In 2008, a clear preponderance of macropterous and dipterous was found, while in 2019 the high share of dipterous remained; however, the number of beetles classified as macropterous decreased in favor of the growth of brachypterous beetles (Figure 9a–c).

The canonical correspondence analysis (CCA) provides the in-depth interpretation of the distribution of particular ecological groups relative to the variables (study years, moisture content) (Figure 10). This analysis showed that the years (2008, 2019) were mainly correlated with the first ordination axis. Moreover, the year 2008, lying in the lower part of the diagram, correlated with the occurrence of xerophilous ground beetles (Xe) foraging on mixed food (Hz). Meanwhile, a separate group clustered around the first ordinance axis, consisting of large beetles (Lz) with moderate moisture requirements (M), typical of forested areas (F) and with the autumnal type of development (Ab), emerged in the second year of the experiment. The analysis of the ordination diagram also implicates the effect of moisture on the formation of carabid assemblages. This environmental variable (moisture content) correlates with the presence of small zoophages (Sz) of the spring type of development (Sb), generalists (G) and beetles preferring open areas (Oa), as well as those carabid species which have higher moisture requirements (We, H) (Figure 10).

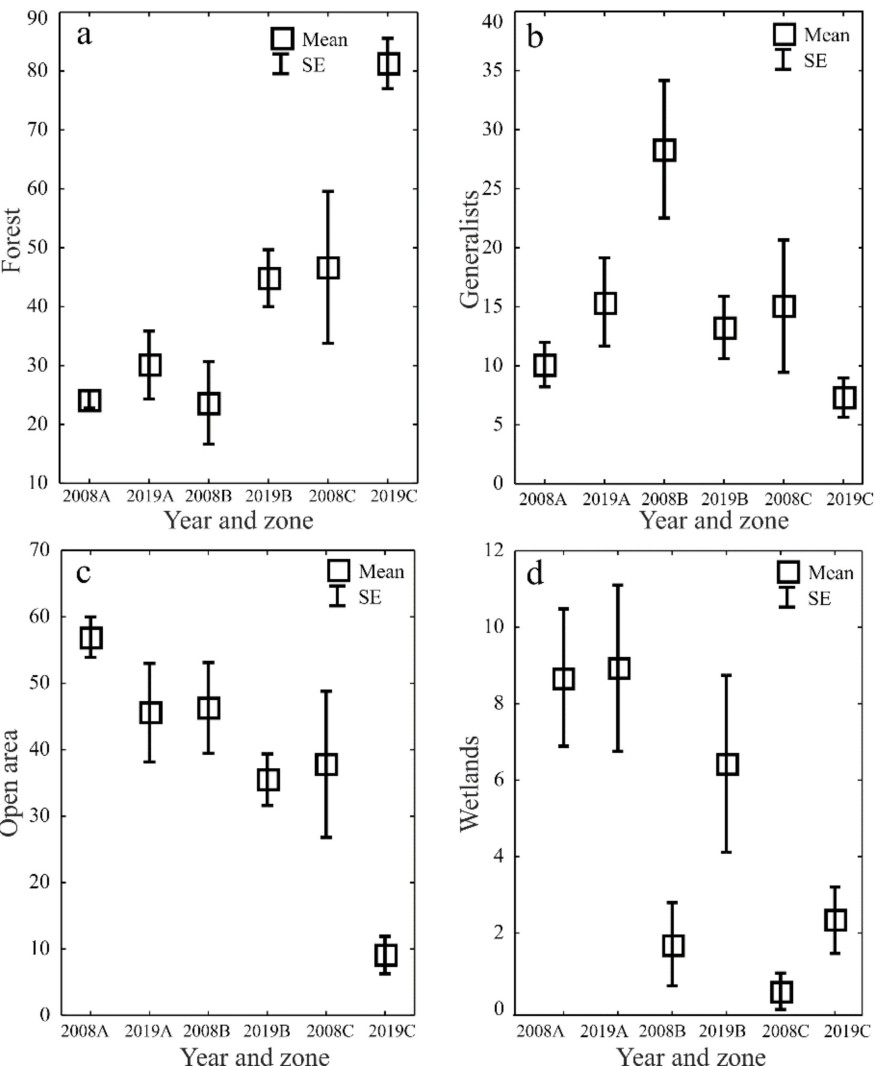

**Figure 5.** Percentages of Carabidae habitat types (forest—(**a**), generalists—(**b**), open area—(**c**) and wetlands—(**d**)) in the analyzed zones (A, B, C) before (2008) and after (2019) the implementation of the small water retention project (vertical lines indicate SE).

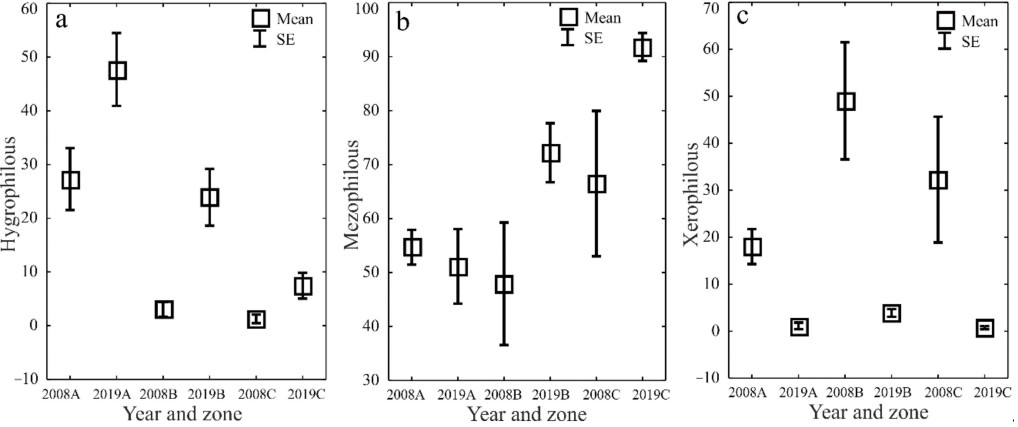

**Figure 6.** Percentages of Carabidae moisture types (hygrophilous—(**a**), mezophilous—(**b**) and xerophilous—(**c**)) in the analyzed zones (A, B, C) before (2008) and after (2019) the implementation of the small water retention project (vertical lines indicate SE).

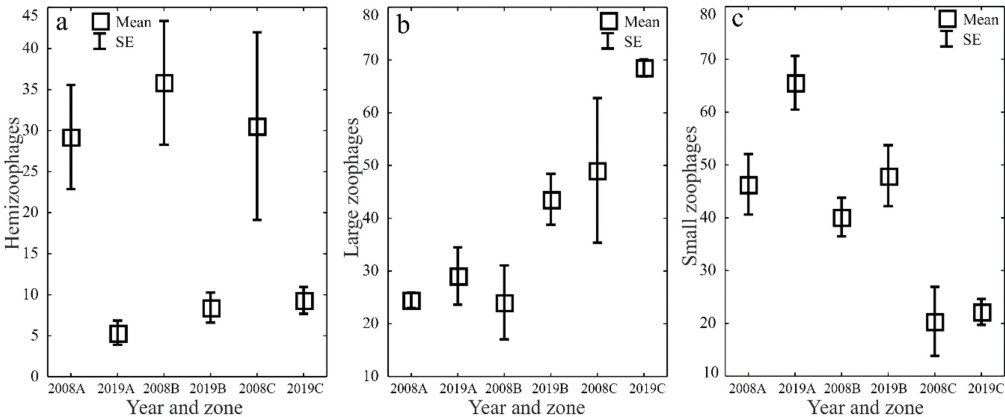

**Figure 7.** Percentages of Carabidae trophic types (hemizoophages—(**a**), large zoophages—(**b**) and small zoophages—(**c**)) in the analyzed zones (A, B, C) before (2008) and after (2019) the implementation of the small water retention project (vertical lines indicate SE).

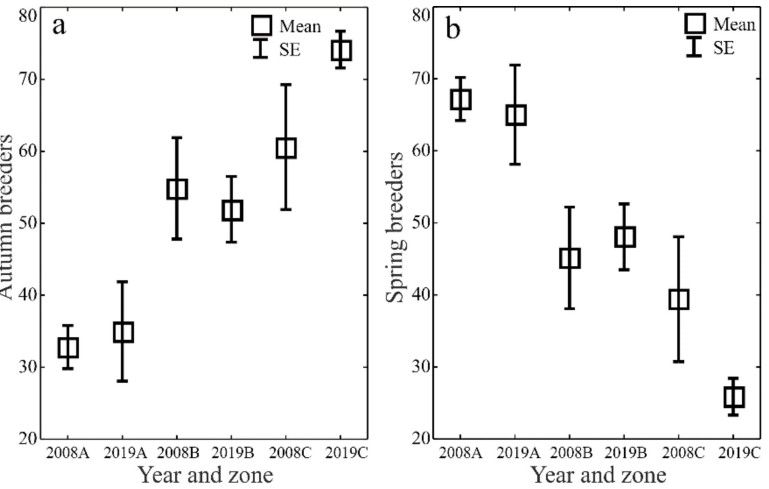

**Figure 8.** Percentages of Carabidae development types (autumn breeders—(**a**) and spring breeders—(**b**)) in the analyzed zones (A, B, C) before (2008) and after (2019) the implementation of the small water retention project (vertical lines indicate SE).

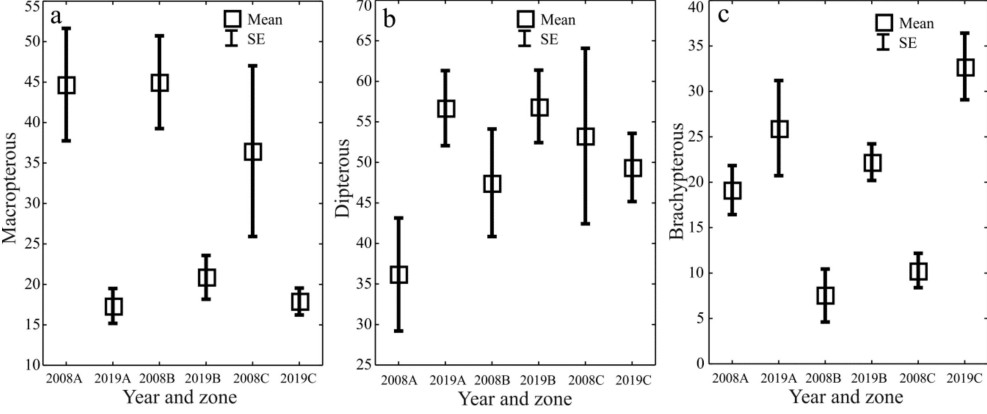

**Figure 9.** Dispersion capability Carabidae (Macropterous (**a**), Dipterous (**b**), Brachypterous (**c**)) in the analyzed zones (A, B, C) before (2008) and after (2019) the implementation of the small water retention project (vertical lines indicate SE).

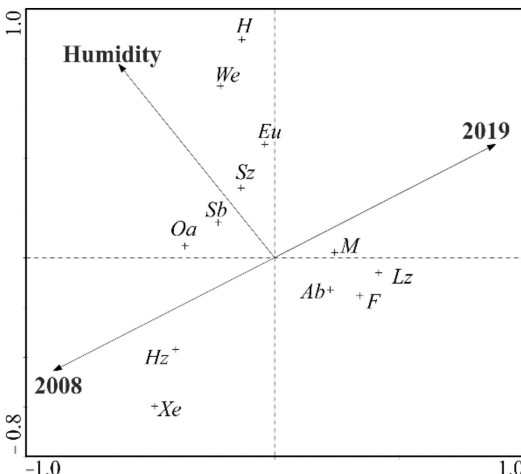

**Figure 10.** Diagram of the canonical correspondence analysis (CCA) presenting the correlations between the Carabidae ecological groups and the selected habitat factors (year, moisture). All abbreviations are as given in the methodology section.

## 4. Discussion

Raising water levels in the environment is an important measure shaping biodiversity, but not every rapid intervention aiming to model the water balance brings about the intended, one-dimensional benefits [7]. The measures were undertaken under the small water retention program to improve water availability. However, they proved insufficient to counteract the decline of biodiversity of ground beetles. An earlier study in the same area [46], which focused on the short-term impact of retention basins on assemblages of ground beetles, showed that shortly after the retention basins had been constructed (in 2012), the number of ground beetles decreased, but the number of species was unchanged. In the following year (2013), a qualitative and quantitative increase was observed among the captured beetles [46]. However, our observations suggest that this increase in biodiversity was possibly only short-lived. After 11 years from our initial entomological observations, a rapid decline in the biological diversity of these insects was noted (Table 2, Figure 3a–c), which agrees with the observed loss of biodiversity worldwide [17,24–27]. The hydro-technical changes improved the environmental conditions in this area by increasing water availability but did not protect the entomofauna from the general decline in biodiversity.

No carabid species under strict protection were found in the analyzed area. However, the presence of three species under partial protection (*Carabus convexus*, *Carabus glabratus*, *Carabus marginalis*) was confirmed, which means that the area submitted to our study plays an essential role in the protection of valuable biodiversity [63]. In 2019, although the water relations had been stabilized, a substantial decrease in the number of specimens of the species *C. marginalis*, with moderate moisture requirements, was observed. Moreover, the transformations of this biocenosis contributed to the disappearance of the highly endangered wetland species *Epaphius rivularis* (Table 2). On the other hand, implementing the water retention project enabled us to detect the presence of *Pterostichus aterrimus*, which is considered a highly hygrophilous species [64]. The construction of small water retention reservoirs in the analyzed area probably affected the share of hygrophilous species characteristic exclusively for peat bogs [53,65]. Once the development project had been completed, their abundance decreased with the simultaneous decline in the number of species representing this group. These findings can indicate that the hydro-technical changes made under the small water retention project induced habitat changes to which a smaller group of hygrophilous species could adapt (Table 2).

According to Czyżyk and Porter [30], the highest number of species and their abundance could be found in a zone located 5 m away from hydro-technical facilities. As observed by these researchers, such a distance enables direct access to water and results in the highest diversity of flora and fauna. However, a reverse relationship was noted during

the entomological observations conducted in the Dąbrówka Forest Subdistrict. Once the retention works had been completed, the highest diversity was recorded along transect C (Figure 3a,b). This zone was dominated by forest species of ground beetles, with the most prevalent species being *P. niger*. The dominance of a single species present in a large number of individuals can suggest a considerable influence of anthropopressure on carabid assemblages. The high share of other specimens from species typical of woodlands is natural and may result from the mid-forest location of the research site.

The NMDS analysis demonstrated the most significant differences in the ground beetles' assemblage in the first year of observations (2008) along transect C (Figure 4). The trees' age most probably had a significant influence on this situation. Young forests, with less compact crowns, are more numerously inhabited by species with low moisture requirements (Xe). The growth of tree crowns and improved water relations following the implementation of the small water retention project resulted in the increased contribution to the structure of ground beetles caught, those representing species with moderate moisture requirements (M) and high moisture requirements (H) and a decrease in the share of xerophilic Carabidae typical for open areas. (Figure 5a–d, Figure 6a–c). This agrees with the studies carried out in pine monoculture forests by Tarwacki [66], who concluded that the dominant share of ground beetles typical of open areas occurred only in the early growth phases of forests.

Czechowski [16] underlines that strong anthropopressure leads to a decrease in the species typical of woodlands. After the small retention basins were created, the number of forest species increased, which may point to the relative stability of this ecosystem despite the intense man-made pressure. However, it is worth bearing in mind that the increase in the number of typical forest species of ground beetles is also a consequence of forest succession. The stability of the habitat in 2019 is also indicated by analysis of the ground beetles ability to fly. In 2019, an increase in the share of brachypterous beetles was observed. This is in line with the research carried out by [67], which found a decline in the share of this group in disturbed habitats.

## 5. Conclusions

The project carried out in the Dąbrówka Forest Subdistrict, whose aim was to improve water relations and, subsequently, achieve greater biological diversity, proved insufficient. The creation of the small water retention system caused a decrease, both quantitative and qualitative, in the captured carabid fauna. However, due to anthropogenic and natural factors, the reduced biological diversity still awaits a more thorough understanding. Furthermore, such transformations are not always negative, as they may be conducive to positive changes in ecologically valuable Carabidae. By improving the availability of water in the habitat, water retention works facilitated the occurrence of specimens from species with specific moisture requirements, often highly valuable in the natural environment. As predicted, the retention works caused changes in ground beetles composition concerning increasing hygrophilous species and a decrease of xerophilous. The rare species *P. aterrimus* appeared, but on the other hand, after the introduction of water retention, we did not observe the presence of the rare peat bog species of *E. rivularis.* Thus, any measures taken to sustain the natural processes that support the struggle against diminishing biodiversity are highly beneficial. All studies that help elucidate the processes that affect biodiversity are of value in reinforcing any policy that supports natural processes.

**Author Contributions:** Validation, formal analysis, software, M.N.; writing, original draft preparation, E.L.; conceptualization, methodology, performed experiments, review and editing, E.L., M.N. and A.K. All authors have read and agreed to the published version of the manuscript.

**Funding:** The results presented in this paper were obtained as a part of comprehensive study financed by the University of Warmia and Mazury in Olsztyn, Faculty of Agriculture and Forestry, Department of Entomology, Phytopathology and Molecular Diagnostics. This work was supported by a research project of the University of Warmia and Mazury in Olsztyn (no. 30.610.010-110).

**Institutional Review Board Statement:** Not applicable.

**Informed Consent Statement:** Not applicable.

**Data Availability Statement:** The data relating to Carabidae harvested in water retention program areas are available upon request to the respective author.

**Acknowledgments:** We would like to thank Olsztyn Forest District for help with material collection. The results presented in this paper were obtained as part of a comprehensive study financed by the University of Warmia and Mazury in Olsztyn, Faculty of Agriculture and Forestry, Department of Entomology, Phytopathology and Molecular Diagnostics. We would like to thank the anonymous reviewers for their valuable comments that helped us improve the manuscript. Thanks to Robert Lee for the linguistic proofreading.

**Conflicts of Interest:** The authors declare no conflict of interest.

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
