# Peer review of "Impact of Water Retention Practices in Forests on the Biodiversity of Ground Beetles (Coleoptera: Carabidae)"

_sustainability, doi:10.3390/su142215068_

Round 1

Reviewer 1 Report

Dear Authors,

Below is my report on the research article titled "Impact of water retention practices in forest on the biodiversity of ground beetles (Coleoptera: Carabidae)".

In this study had carried out in an area covered by the "Increasing retention capacity and prevention of floods and droughts in forest ecosystems in lowland areas" land drainage development project.

The aim of this study had evaluated the impact of transformations in a habitat following the project’s implementation on the structure of assemblages of epigeic ground beetles, in the context of the overall trend of decreasing biodiversity.

The entomological observations had commenced before launching the planned hydro-technical development at the study area, and were repeated 11 years later.

During 2008 and 2019 years of observations 3,311 specimens of ground beetles, representing 89 species had captured.

Water regulation  had affected the composition of ground beetle assemblages.

After the small water retention program had been completed, a quantitative and qualitative decrease in ground beetles had observed.

The groundwork for the development caused some disturbances in the habitat, leading, amongst others, to higher soil moisture, which had conducive to the establishment of ground beetle assemblages other than those observed before the water retention project.

After the hydro-technical objects had been constructed, the share of large and small zoophages with higher moisture preferences (hygrophilous species) increased, while the contribution of xerophilous hemizoophages had decreased.

This manuscript is contain sufficient original and new scientific data.

Separately, introduction, materials and methods, results, discussion and reference list are sufficient. Separately, statistical analysis was made appropriately.

This manuscipt can be publish in this journal.

Best regards

Author Response

Thank you very much for evaluating our manuscript and making it available for publication in this journal.

Reviewer 2 Report

The paper presents a very valuable ecological and faunal study of the insect group Carabidae and undoubtedly deserves to be published. However, the authors are advised to reflect and respond to the following comments:  

  1. It seems that in the “introduction” the authors should mention similar studies they had already previously conducted on the ground beetle fauna in the DÄ…brówka Forest Subdistrict in 2009, 2012 and 2013 years.
Although the authors cite a relevant paper from 2020 in the literature [62],
they only refer to it in the discussion.
2. The study of the entomofauna of Carabidae at DÄ…brówka Forest Subdistrict   was conducted on the one site - “study site 1” over 5 years: in 2008 and 2019 (data presented in the present manuscript) and in 2009, 2012 and 2013 (data presented in the
publication from Ludwiczak et al. 2020). That studies were very intensive
in that: i. it lasted for the entire growing season (from April/May to October/November), ii. glycol traps were used (adult insects together with probably larvae were killed), iii. the traps were replaced every 14 days.
The authors in the manuscript compare the Carabidae population in two periods: 2008 (before the hydro-technical changes) and 2019 (after the hydro-technical changes), but they do not mention that insects were trapped
for 3 seasons in between (2009, 2012, 2013). It would have to be ruled out, however, that the trapping of beetles for these three years did not directly affect the decline of ground beetle populations in the study area. Moreover, comparing the abundance of selected 13 species (e.g. Harpalus tardus, Harpalus rufipes, Amara aenea) at site 1, in 2008 and 2009 (from Ludwiczak 2020), it was found: in 2008 the species were abundant or very abundant, in 2009 the abundance of most species (13 species) decreased significantly or very significantly (even by 90%). How can this correlation
be explained? In conclusion, the Authors are asked to answer the question:
could the very intensive ground beetle trapping carried out for 4 years at
one site (site 1) have had a direct, negative impact on the insect abundance
in 2019?

Moreover:

1. Page 5. The statement “In the first research year (2008), carabids were captured from 20 April to 3 November (197 days), whereas in 2019, the entomological observations were carried out from 10 May to 22 November (196 days). The research was delayed in 2019 due to the low temperatures in April” seems incomprehensible.
According to weather archive data for 2008 and 2019: In 2019, temperatures in Olsztyn after 20 April were quite high, reaching up to 26 C. In 2008 temperatures were considerably lower, as did not exceed 20 deg C. A similar trend was recorded for minimum temperatures in April 2008 and 2019. Therefore, postponing the start of the field studies in 2019 due to the temperature conditions seems unjustified. The delay in the start of the observations by about 20 days may have had some negative impact on the results and the abundance of beetles collected in the early spring period. The high air temperatures during this period in 2019 certainly intensified the activity of the insects, which left the overwintering site (where they often congregate in large numbers) and dispersed.

2. Page 13. Line 349 and next.

After 11 years from our initial entomological observations, a rapid decline in the biological diversity of these insects was noted (Table 2, Fig- 350 ure 3 A-C), which agrees with the global trends”.

Please specify: what global trends the authors have in mind? Citations missing.

3. Page 13. Line 351-353.

“The hydro-technical changes improved the environmental conditions in this area but did not protect the entomofauna from the general decline in biodiversity”.

Please specify: how the “changes improved the environmental conditions”,

4. Page 14. Line 363-and next.

The construction of small water retention reservoirs in the analyzed area probably affected the share of hygrophilous species characteristic exclusively for peat bogs (tyrophobionts and tyrophiles) [52,65]. Once the development project had been completed, their abundance decreased with the simultaneous decline in the number of species representing this group. These findings can indicate that the hydro-technical changes made under the small water retention project induced habitat changes to which a smaller group of hygrophilous species could adapt (Table 2).”

Please specify: which group of species are tyrophobionts or tyrophiles,there are no data/information about such group in “results”, how much their abundance decreased etc.

5. Page 14. Line 371-377 “According to Czyżyk and Porter [30], the highest number of species and their abundance could be found in a zone located 5 meters away from hydro-technical facilities. As observed by these researchers, such a distance enables direct access to water and results in the highest diversity of flora and fauna. However, a reverse relationship was noted during the entomological observations conducted in the DÄ…brówka Forest Subdistrict. Once the retention works had been completed, the highest diversity was recorded along transect C (Figure 3 A, B)”.

Please check once more the results and correctness of Figure 3 B (Page 9). It can be seen from Table 2, in 2019, on site 1 and 2 in transect B and A, respectively, species diversity was the highest (higher than in transect C).

6. Page 14. Line 377-380

“This zone was dominated by forest species of ground beetles, with the most prevalent species being P. niger. The dominance of a single species present in a large number of individuals can suggest a considerable influence of anthropopressure on carabid assemblages”.

Please note, that this species was also a dominant before environmental
changes, in 2008

Author Response

  1. Changes have been made in the introduction.
  2. The reviewer noted that the observations were conducted in 2008 and compared these data to the 2009 observations published by us in PeerJ. The comparison shows that in 2008 more individuals were caught (2,085) compared to 2009 (1,377). The reviewer also noted that "in 2008 the species were abundant or very abundant, in 2009 the abundance of most species (13 species) decreased significantly or very significantly (even by 90%)". Our research goal was not to compare the two years of research. Data from 2008 and 2009 came from the same area, but the location of these traps was not identical. Hence, probably the number of species caught varies. Referring to the reviewer's remark on the changes in the composition of dominant species (the reviewer mentioned 13 species), 10 species in both years of the study (2008 and 2009) were identical (Tab. I).

    Referring to the remark "could the very intensive ground beetle trapping carried out for 4 years at one site (site 1) have had a direct, negative impact on the insect abundance in 2019?" When analyzing the number of Carabidae in the subsequent years of research in the study area, we find a decrease in the number of Carabidae individuals caught. The question posed by the reviewer is therefore absolutely justified. However, did catching Carabidae in the years of research affect the Carabidae abundance in 2019? We believe not. Most of the Carabidae in the Central European zone are one-year populations, and catching a small part of the population during the study is not a factor that is mentioned as significantly affecting the size of this group in the following years. Referring to the results of our publication (PeerJ) cited by the reviewer, in the position I in 2012 927 individuals were caught, and a year later in the same place 1493, i.e. more. The research may seem intense as the material was collected over the following seasons, however, the number of traps set was not high, but sufficient to reach the estimated number of species from a given area, as confirmed by the rarefaction curves.

    Table I

    Comparison of the most numerous species caught in DÄ…brówka in 2008 and 2009

    The presented manuscript

    Ludwiczak E., Nietupski M., Kosewska A. 2020. Ground beetles (Coleoptera; Carabidae) as an indicator of ongoing changes in forest habitats due to increased water retention. PeerJ 8:e9815 https://doi.org/10.7717/peerj.9815

    2008

    2009

    Species

    Number

    Species

    Number

    Pterostichus niger (Schaller, 1783)

    424

    Pterostichus niger (Schaller, 1783)

    305

    Calathus erratus (C.Sahlberg, 1827)

    257

    Calathus erratus (C.Sahlberg, 1827)

    237

    Harpalus tardus (Panzer, 1796)

    256

    Epaphius secalis (Paykull, 1790)

    125

    Pterostichus vernalis (Panzer, 1796)

    152

    Poecilus versicolor

    96

    Carabus marginalis Fabricius, 1794

    101

    Dyschirius globosus (Herbst, 1784)

    84

    Dyschirius globosus (Herbst, 1784)

    95

    Bembidion gilvipes (Sturm, 1825)

    78

    Harpalus rufipes (De Geer, 1774)

    73

    Harpalus tardus (Panzer, 1796)

    78

    Epaphius secalis (Paykull, 1790)

    65

    Carabus marginalis Fabricius, 1794

    63

    Amara aenea (De Geer, 1774)

    63

    Pterostichus diligens (Sturm, 1824)

    34

    Harpalus rubripes (Duftschmid, 1812)

    53

    Agonum fuliginosum (Panzer, 1809)

    32

    Amara lunicollis Schiodte, 1837

    47

    Harpalus rufipes (De Geer, 1774)

    28

    Amara communis (Panzer, 1797)

    44

    Amara lunicollis Schiodte, 1837

    26

    Bembidion gilvipes (Sturm, 1825)

    43

    Amara communis (Panzer, 1797)

    18

    Moreover:

    1. 

    The research conducted in 2019 in the area of DÄ…brówka Wielka started with a delay compared to 2008, this is our mistake. We only considered the duration of the experiment (2008-197 days; 2019-196 days) and not the compliance with the study start dates. Entomological observations in the area covered by the "Small Retention Program" have been carried out since 2008 and their continuation is also planned in the coming years. Conducting many years of field research at exactly the same time is, however, difficult, many times even impossible because it depends on a number of factors. Meteorological conditions are one of the most important factors determining the starting and ending of entomological research. There is no meteorological station in the area of our research, from which we could obtain accurate data on meteorological conditions in this area. The nearest weather station is 15 km away, in Olsztyn, therefore the decision to start research is always a subjective assessment of the local conditions of the observers. The beginning of field research is also related to their technical organization, which often interferes with the other research or teaching duties of the people conducting the research.

    2. Changes have been made. Citations missing added.
  3. Changes have been made.

  4. We didn’t provide this division in the methodology because we did not analyze it. We only state that there is such a division. It may be unclear, so we have removed it (tyrophobionts and tyrophiles).

  5. Figure 3B presents the results of statistical analyzes and the values of the average abundance of species in the analyzed combinations in the years of the study.This makes it possible to compare the values of, for example, species richness, based on statistically significant differences.In this approach, each trap was treated as a sample.In the table, the values are for the actual number of speciescaught during the research.

  6. This species dominated in both years of observation. This information was highlighted in the discussion of the results ("Pterostichus niger, a woodland species with moderate moisture requirements, was the prevalent species in both years").

Reviewer 3 Report

Study is very interesting an gave important information about the effect of water regulation in composition of ground beetles, considering a lon period of time between comparison. Only the classificiation of different carabid groups is no so clear, author must be explain it better and improve the conclusions. Authors can considered estimate effective number of species and also species turnover between sample periods. 

Author Response

Referring to the remark „the classification of different carabid groups is not so clear classification of different carabid groups is described in the methodology. The division of ecological groups used in the manuscript was made on the basis of commonly available studies (Lindroth, 1986; Thiele, 1977; LeÅ›niak, 1985; Aleksandrowicz, 2004). If it is unclear, please let me know which division should be discussed in more detail, and we will be happy to refine it.

Regarding the reviewer's remark “Authors can consider estimate effective number of species and also species turnover between sample periods”: To assess the adequacy of the sampling efficiency we used the Jackknife 2 estimator (program EstimateS) and we obtained the species accumulation curves presented in the figure.

The requests have been corrected.

Linguistically, the manuscript was checked by a native speaker Robert Lee.

Reviewer 4 Report

The study is not very original and the results are quite obvious.
In any case, the study is conducted with sufficient methodological rigor, statistical data processing is sufficient and the conclusions, however skimpy, are logical and consequential to the results.

Author Response

Thank you very much for reviewing our manuscript. The conclusions were extened.

Round 2

Reviewer 2 Report

Dear Authors; Thank you for your clarification.